# Predictive Biomarkers for Immune-Checkpoint Inhibitor Treatment Response in Patients with Hepatocellular Carcinoma

**DOI:** 10.3390/ijms24087640

**Published:** 2023-04-21

**Authors:** Jun Ho Ji, Sang Yun Ha, Danbi Lee, Kamya Sankar, Ekaterina K. Koltsova, Ghassan K. Abou-Alfa, Ju Dong Yang

**Affiliations:** 1Division of Hematology and Oncology, Department of Internal Medicine, Samsung Changwon Hospital, Sungkyunkwan University School of Medicine, Changwon 51353, Republic of Korea; 2Karsh Division of Gastroenterology and Hepatology, Comprehensive Transplant Center, Samuel Oschin Comprehensive Cancer Institute, Cedars-Sinai Medical Center, Los Angeles, CA 90048, USA; 3Department of Pathology and Translational Genomics, Samsung Medical Center, Sungkyunkwan University School of Medicine, Seoul 03181, Republic of Korea; 4Department of Gastroenterology, Liver Center, Asan Medical Center, University of Ulsan College of Medicine, Seoul 05505, Republic of Korea; 5Division of Medical Oncology, Samuel Oschin Comprehensive Cancer Institute, Cedars-Sinai Medical Center, Los Angeles, CA 90048, USA; 6Department of Medicine, Samuel Oschin Comprehensive Cancer Institute, Smidt Heart Institute, Cedars-Sinai Medical Center, Los Angeles, CA 90048, USA; 7Memorial Sloan Kettering Cancer Center, New York, NY 10065, USA; 8Weil Cornell Medicine, Cornell University, New York, NY 14853, USA

**Keywords:** Biomarker, Hepatocellular carcinoma, Immune checkpoint inhibitor

## Abstract

Hepatocellular carcinoma (HCC) has one of the highest mortality rates among solid cancers. Late diagnosis and a lack of efficacious treatment options contribute to the dismal prognosis of HCC. Immune checkpoint inhibitor (ICI)-based immunotherapy has presented a new milestone in the treatment of cancer. Immunotherapy has yielded remarkable treatment responses in a range of cancer types including HCC. Based on the therapeutic effect of ICI alone (programmed cell death (PD)-1/programmed death-ligand1 (PD-L)1 antibody), investigators have developed combined ICI therapies including ICI + ICI, ICI + tyrosine kinase inhibitor (TKI), and ICI + locoregional treatment or novel immunotherapy. Although these regimens have demonstrated increasing treatment efficacy with the addition of novel drugs, the development of biomarkers to predict toxicity and treatment response in patients receiving ICI is in urgent need. PD-L1 expression in tumor cells received the most attention in early studies among various predictive biomarkers. However, PD-L1 expression alone has limited utility as a predictive biomarker in HCC. Accordingly, subsequent studies have evaluated the utility of tumor mutational burden (TMB), gene signatures, and multiplex immunohistochemistry (IHC) as predictive biomarkers. In this review, we aim to discuss the current state of immunotherapy for HCC, the results of the predictive biomarker studies, and future direction.

## 1. Introduction

Hepatocellular carcinoma (HCC) is the most common primary liver cancer and the third leading cause of cancer death worldwide, with 906,000 new cases and 830,000 HCC-related deaths annually [1]. Repeated necrosis and regeneration of hepatocytes caused by chronic inflammation and injury gradually progress to liver fibrosis and cirrhosis, eventually leading to HCC. Unfortunately, most patients with liver cirrhosis are asymptomatic, so HCC is often diagnosed at advanced stages. Accordingly, the mortality of HCC is high and continues to rise [2]. Patients with advanced-stage HCC may benefit from systemic therapy and, until recently, most systemic treatments for HCC comprised targeted therapy with tyrosine kinase inhibitors (TKI) such as sorafenib, lenvatinib, regorafenib, and cabozantinib [3]. 

The immune system can recognize foreign cells based on the proteins present on the cell surface and has the ability to eliminate different from our own body such as viruses, bacteria, and malignancies. In this process, checkpoint proteins are to limit autoimmune damage to normal tissue by preventing T-cell activation. HCC and other tumors use this mechanism to evade immune responses by expressing ligands on the tumor cell surface. In addition, tumor cells promote immune evasion by interfering with the recognition of tumor antigen presentation or generating the immunosuppressive tumor microenvironment [4]. Immune checkpoint inhibitors block the interaction between checkpoint proteins and their ligands, thereby preventing the inactivation of T cell function. Chronic viral infections such as hepatitis B and hepatitis C, which are the main causes of HCC, promote chronic inflammation of the liver, and in patients with chronic inflammatory liver disease, PD-1 overexpression in lymphocyte and PD-L1/PD-L2 overexpression in stromal cells (Kupffer cell, liver sinusoidal endothelial cells) are observed. This upregulation of checkpoint proteins suggests that ICIs may be effective in HCC. The use of immunotherapy alone or in combination with targeted agents results in improved survival with a durable response and has become the new standard of care with several successful phase 3 studies published since 2020 [5,6,7]. However, despite these inspiring results, since HCC is a heterogeneous disease with diverse immunological characteristics, immunotherapy does not guarantee a clinical benefit in all patients with HCC, and more than two-thirds of advanced cancer patients do not respond to immunotherapy [8,9]. Moreover, immunotherapy improved long-term survival in cancer treatment, but considering the trend of crossing the survival curve in randomized clinical trials for ICIs, it suggests that the early mortality rate is rather higher in the ICI treatment group compared to the control group [10]. To overcome these limitations and optimize the use of ICIs in HCC, the development of predictive biomarkers that can be used to identify individuals who are more likely to experience favorable or unfavorable effects of immunotherapy has become increasingly important [11,12]. For biomarker discovery, testing of blood or feces can be used to obtain complex data using advanced technologies such as genomics, proteomics, metabolomics, and artificial intelligence. Unfortunately, the need for the development of clinically applicable predictive biomarkers remains unmet despite efforts to identify biomarkers predicting outcomes of immunotherapy for HCC. 

Herein, we aim to discuss predictive biomarkers for immunotherapy response in HCC, with a focus on clinical applications (Figure 1.)

## 2. Biomarkers Related to the Interaction between T Cells and Tumor Cells

### 2.1. DNA Damage Repair Pathway (dMMR/MSI-H)

DNA damage repair (DDR) genes play key roles in maintaining the stability of the human genome, and disruption of DNA damage repair pathways by germline or somatic mutations in DDR genes may contribute to the development of cancer [13,14]. The major DNA repair mechanisms are mismatch repair (MMR), homologous recombination (HR), polymerase proofreading (*POLE/POLD1*), base excision repair (BER), nucleotide excision repair, and DNA repair mediated by MGMT (O^6^-methylguanine–DNA methyltransferase) [15]. Disruptions to BER, HR, and MMR contribute more significantly to tumor mutational burden (TMB) or neoantigens, which have the highest levels when co-mutated. [16] In HCC, these DDR alterations showed distinct characteristics. Peng Lin et al. [17] categorized DDR alterations in HCC into two heterogeneous subtypes (the DDR-activated subtype and the DDR-suppressed subtype) and compared prognosis and clinicopathologic features between subtypes. HCC patients with the DDR-suppressed subtype tended to have longer survival. Significant activation of CD4+ T cells, central memory CD4+ T Cells, and effector memory CD4+ T cells was observed in HCC patients with the DDR-activated subtype. The authors also evaluated the relationships between DDR signatures and immunotherapy responses in an external cohort, with a high DDR subtype score observed in patients with complete response (CR) and partial response (PR) [18]. In a separate Chinese study, [19] DDR pathway/gene mutations were evaluated using next-generation sequencing (NGS) analysis in 1427 HCC patients who had undergone surgery. Among the included patients, 18.8% carried somatic mutations in DDR genes and 3.5% carried germline mutations in DDR genes. TMB was significantly higher in HCC patients who carried somatic mutations in DDR genes, while no difference in TMB was observed in HCC patients with DDR germline mutations. The results of this study indicate that somatic mutations of DDR genes contribute to high TMB, suggesting a greater response to immunotherapy in patients with DDR gene mutations. In addition, each DDR mechanism has been studied for ICI response prediction, and in particular, the MMR pathway is one of the most well-known biomarkers for cancer immunotherapy. DNA MMR maintains genomic stability by repairing mismatched bases or mispairs during DNA replication and recombination, thereby preventing genetic alteration [20]. Genetic mutations involving the DNA MMR pathway are associated with increased microsatellite instability (MSI), high numbers of somatic mutations, and a higher number of tumor-infiltrating lymphocytes. This process promotes an anti-tumor cytotoxic immune response and secretion of soluble factors that contribute to the activation of the programmed death ligand-1 (PD-L1) pathway within the tumor microenvironment [21,22]. dMMR is also associated with DNA polymerase gene epsilon/delta1 (*POLE/POLD1*) mutations, with increased mutation, and neoantigen load. In an analysis of 47,721 patients with solid cancers, *POLE/POLD1* mutations were associated with TMB-H. dMMR, TMB-H, and high neoantigen loads are thought to be associated with immunotherapy response [23]. The dramatic response of ICI shown by patients in the dMMR/MSI-H group in other tumors led to expectations that those of dMMR/MSI-H in HCC would play a similar role, but contrary to expectation, dMMR/MSI-H in HCC is limited in its use as a predictive biomarker for immunotherapy due to their low prevalence of less of than 3% [24,25,26,27]. MSI status is considered a potential biomarker for response to immunotherapy. In 2015, the FDA granted accelerated approval for the use of pembrolizumab for unresectable solid tumors with MSI-H or dMMR based on the results of the KEYNOTE-016 study. Unfortunately, since less than 3% of patients with HCC have high MSI status, the utility of assessing MSI status in HCC is expected to be limited [25,27,28,29,30,31,32].

### 2.2. Tumor Antigen Presentation

#### 2.2.1. PD-L1/PD-1 Expression 

PD-1, identified by Honjo et al. [33] is expressed on the surface of T lymphocytes and is a master regulator of immune cell tolerance. The binding of PD-L1 to PD-1 inhibits T cell effector functions [34]. PD-L1 was identified by Honjo et al. as a PD-1 ligand in 2000 [35]. The following year, the expression of PD-1 ligands on tumor cell lines was demonstrated, with the blockade of PD-L1 shown to exert an anti-tumor effect [36]. PD-L1 is expressed by a range of tissues. The binding of PD-L1 expressed on tumor cells and virus-infected cells to PD-1 expressed on T cells leads to direct inhibition of T cell proliferation and effector functions such as IFN-γ production and cytotoxic activity [37]. 

In the liver, PD-L1 is expressed by Kupffer cells, hepatocytes, and hepatic sinusoidal cells [38]. PD-L1 expression is observed in 10–20% of HCC samples. PD-L1 overexpression in HCC has been reported to be associated with poor prognosis [39,40]. Zhou et al. [41] identified suppressive immune checkpoint molecules in patients with early HCC and highlighted the importance of PD-1 in suppressing the function of tumor-infiltrating lymphocytes.

However, the utility of PD-L1 expression as a predictive biomarker in remains controversial. The phase I/II CHECKMATE 040 study [42] investigated the role of nivolumab in HCC patients that failed to respond to sorafenib. In this study, the objective response rate (ORR) was 19% in patients with <1% PD-L1 expression in tumor biopsy samples and 26% in patients with ≥1% PD-L1 expression in tumor biopsy samples using a PD-L1 IHC 28-8 pharmDx assay; however, this difference did not reach statistical significance. In a retrospective exploratory analysis of the results of the KEYNOTE 224 trial [43] on the effect of second-line pembrolizumab in HCC, baseline PD-L1 expression on Tumor cells (TC) also did not affect response rates to pembrolizumab treatment. However, the baseline combined positive score (CPS) based on PD-L1 expression on tumor cells and immune cells (macrophages and lymphocytes) was associated with response to pembrolizumab treatment. In the IMbrave150 study [44], PD-L1 expression measured using the SP263 assay had no significant effect on immunotherapy in patients with a CPS < 1% but could predict a good effect of atezolizumab/bevacizumab treatment compared to sorafenib treatment in patients with a CPS ≥ 1% (HR, 0.52; 95% confidence interval [CI], 0.32–0.87). In the HIMALAYA study, [45] PD-L1 expression measured using the SP263 assay was not significantly correlated with the therapeutic efficacy of durvalumab monotherapy or durvalumab and tremelimumab combination therapy. 

Recent prospective studies in HCC have demonstrated that the use of PD-L1 expression alone has limited accuracy in predicting the efficacy of immune checkpoint inhibitors. Moreover, the antibody assays used for immunohistochemical studies, and the cut-off values used for determining PD-L1 positivity and PD-L1 expressing cells differ between studies of PD-L1. Standardization of PD-L1 analyses is urgently required in future studies.

Circulating soluble PD-L1 levels are negatively correlated with ICI efficacy in patients with NSCLC and melanoma [46,47,48,49]. In HCC, several studies [50,51,52] have reported an association between soluble PD-L1 levels and prognosis, with high serum PD-L1 levels indicating a poor prognosis. Tissue sampling of HCC is not always feasible. Moreover, it is not easy to obtain adequate amounts of tissue for reliable genetic studies. In clinical practice, there is increasing interest in the measurements of soluble PD-L1 due to the advantages of being non-invasive and enabling repeated sampling. More clinical evidence is needed for PD-L1 using liquid biopsy in HCC. 

#### 2.2.2. TMB

TMB has been regarded as a predictive biomarker for the response to immunotherapy. A high TMB is variably defined as tumors with ≥17 or ≥20 mutations/Mb upon sequencing of at least 1.2–1.5 Mb of the tumor genome [53]. High TMB is reportedly associated with increased neoantigen load [54]. Neoantigens contribute to the immune recognition of tumor cells and the subsequent induction of anti-tumor responses, which can be boosted by the use of immune checkpoint inhibitors [55]. There is a wealth of data that demonstrated that TMB is associated with immunotherapy response, with TMB having predictive value as a biomarker in most cancer subtypes [56]. As a TMB-H is prevalent in melanoma [57] and lung cancer [58], which are closely related to exposure to mutagens such as smoking and ultraviolet light, [59] studies on the efficacy of immunotherapy were first conducted in patients with melanoma or lung cancer. 

While the role of TMB as a predictive marker of immunotherapy response in patients with HCC remains to be elucidated, in a large-scale study [56] conducted on 27 tumor types, TMB was correlated with ORR to anti-PD1 therapy, with a median TMB of 4–5 mutations/Mb in the 43 patients with HCC (21 treated with nivolumab and 22 treated with durvalumab). In a separate large-scale study of 24 tumor types conducted in China, [60] the median TMB in patients with HCC was reportedly 4–5 mutations per Mb. Ang et al. reported only 0.8% (n = 6) of 755 patients with HCC had TMB-H status when a TMB-H cut-off value of 4 mutations/Mb was used. In this study, TMB-H status was not associated with response to ICI therapy, likely due to the small sample size. In a phase Ib study [61] of camrelizumab and afatinib comprising 16 HCC patients, TMB values were correlated with treatment response. An exploratory analysis of the GO30140 study demonstrated that the ORR was higher in patients with TMB-H status, while TMB was unable to predict the treatment response or progression-free survival (PFS) in patients with HCC treated with atezolizumab plus bevacizumab. In the IMbrave150 study, no correlation was observed between TMB and ORR or PFS [62]. 

Several factors can influence the results of TMB analysis including the sequencing panel, type of mutations, and cut-off points used to define high TMB. WGS (whole genome sequencing) is the gold standard method of TMB analysis; [63] however, WES (whole exome sequencing) predominantly using NGS panels is routinely used for TMB analysis in clinical practice due to lower cost and time requirements [64]. TMB assays are not standardized due to the use of different sequencing platforms, calculations, and cut-off values. The use of TMB as a reliable biomarker is limited by the strict technical specifications, time, and high costs. Additionally, the low proportion of patients with HCC that have TMB-H status decreases the efficiency of TMB evaluations. Moreover, not all patients with TMB-H status respond to immune checkpoint inhibitors. Mcgrail et al. demonstrated that TMB-H was not associated with CD8+ T cells or neoantigen loads in breast cancer, prostate cancer, and gliomas. In these patients, the ORR of immunotherapy was less than 20%. This result was similar in patients with TMB-L [64]. Cristescu et al. conducted a study [65] based on the KEYNOTE clinical dataset and found that the response rate to pembrolizumab was higher in patients with both TMB-H and T cell inflamed GEP-H (gene express pattern, high) statuses (37–57%) than patients with TMB-H alone (11–42%). In this study, less than 5% of patients with HCC had TMB-H and GEP-H status, which were associated with favorable prognoses. Taken together, these results indicate that TMB alone may not be an ideal biomarker for predicting treatment response to ICIs. 

Finally, TMB does not appear to be associated with PD-L1/PD-1 expression. Although there are reported differences between tumor subtypes, the concordance of PD-1/PD-L1 expression and TMB-H is reportedly 32% in melanoma, 12% in non-small cell lung cancer (NSCLC), 2.4% in endometrial cancer, 2.2% in esophageal cancer, and 1.2% of colorectal cancer. For all cancer types, a low proportion of patients had both TMB-H and PD-L1 expression. [53] Given these differences in TMB status and other biomarkers between cancer subtypes, further comprehensive studies are required to determine the utility of TMB and other biomarkers in predicting responses to immunotherapy in patients with HCC. 

#### 2.2.3. Interferon-Gamma Signaling Pathway

Interferon-gamma (IFN-γ) is released by T cells activated in response to recognizing neoantigens. The binding of IFN-γ to INF-γ receptors on tumor cell membranes activates Janus Kinase 1 (JAK1) and JAK2, which are signal transducers and activators of transcription (STAT) signaling that promote the expression of IFN-related genes, including interferon regulatory factor 1 (*IRF1*). Expression of *IRF1* induces the transcription of other genes that increase surface expression of PD-L1 and MHC molecules.

Gao et al. [66] demonstrated that loss of INF-γ signaling is associated with primary resistance to anti-cytotoxic T-lymphocyte associated protein-4 (CTLA-4) therapy, with an average of 15.33 mutations in genes related to the IFN-γ pathway in non-responders compared to an average of only one mutation in responders. Of the 12 melanoma tumors with primary resistance to ipilimumab, 75% (9/12) harbored circulating nucleic acids (CNAs) related to INF-γ pathway genes (*IFNGR1*, *IFNGR2*, *IRF1*, *JAK2,* suppressor of cytokine signaling 1 [*SOCS1*], and *STAT4*), while tumors from responders did not harbor CNAs related to INF-γ signaling (0/4, 0%). INF-γ related gene signatures have been studied as a potential biomarker for predicting response to ICIs in various tumor types including melanoma, NSCLC, and urothelial carcinoma (UCC) [67,68,69,70,71]. A study published in 2000 [72] reported that IFN-γ receptor expression in HCC was associated with escape from host immune surveillance, indicating that the IFN-γ pathway could play an important role in the immune evasion of tumors. Numata et al. confirmed that IFN-γ and interleukin 1β (IL-1β) exert synergistic effects on PD-L1 expression in HCC cells. A Chinese study identified IRF-8, one of the nine IRF members that function in regulating the IFN-γ pathway, as a potential biomarker for predicting response to anti-PD-1 treatment in HCC [73].

The JAK/STAT pathway has also been reported to be associated with resistance to ICIs. Loss of function mutations in JAK1/2 can lead to acquired and primary resistance to anti-PD-1 therapy [74,75]. The suppressor of the cytokine signaling protein (SOCS) family is a group of intracellular proteins that generally function as inhibitors of the IFN-γ signaling pathway. SOCS protein has been shown to negatively regulate tumor cell proliferation, differentiation, and immune response in HCC [76]. Identification of JAK1/2 and SOCS mutations before and during ICI therapy can help predict treatment courses and outcomes. Accordingly, gene signatures related to the IFN-γ signaling pathway may have utility as biomarkers for predicting response to immunotherapy in HCC; however, further studies are required are needed in the future. 

#### 2.2.4. Alterations in Genes Related to Immune Function

Wnt signaling is associated with immune surveillance escape, and B-catenin is known to play an important role in HCC survival by promoting EGFR signaling in the early phase of carcinogenesis [77]. Deregulation of WNT/B-catenin signaling was reportedly observed in 40–80% of patients with HCC in a preclinical study. Tumors harboring WNT/B-catenin aberrations are reportedly resistant to immunotherapy in mouse models [78,79]. James et al. identified more than 341 cancer-associated genes in 127 patients with HCC using prospective NGS. In this study, [80] Wnt/β-catenin mutations were reported to be significantly associated with resistance to immunotherapy, with a shorter median overall survival (OS, 9.1 vs. 15.2 months) and PFS (2.0 months vs. 7.4 months) compared to patients with wild-type tumors. 

Dai et al. [81] posited that immune-related gene signatures have utility as predictive biomarkers of ICI efficacy in HCC and identified 11 immune-related genes through analysis of 365 HCC-samples to create the IRGPI (immune-related gene-based prognostic index). The IRGPI was shown to have utility in identifying patients with HCC who are immunogenic and more sensitive to immunotherapy. Genes related to Wnt/B-catenin signaling, a widely studied pathway in HCC, may represent good candidates for predictive biomarkers of immunotherapy.

*TP53* gene aberrations, independently of *CTNNB1* gene aberrations, are associated with specific IFN-γ gene signatures, higher Foxp3+ Treg infiltration, and lower CD8+ T cell infiltration in HCC [82,83,84]. Wang et al. reported that 61.8% of a Chinese HCC cohort (n = 369) had *TP53* gene mutations, and *TP53* mutation was associated with TMB-H and worse survival in HCC [85]. Accordingly, immune cell infiltration of tumor tissues, gene signatures, TMB-H, neoantigens, and DDR gene mutations all appear to be associated with poor ICI response. Studies on the association between *TP53* gene mutations and ICI response have been conducted in several types of carcinomas [86,87,88]. *KRAS* is involved in the development of many human cancers. The prevalence of *KRAS* mutations is less than 20% in HCC tissues [89,90,91] and the contribution of *KRAS* mutations to the pathogenesis of HCC remains unclear. Although most previous studies of *KRAS* mutations have been conducted in patients with NSCLC, previous studies have reported conflicting results regarding the association between *KRAS* mutations and the efficacy of immunotherapy [92,93,94]. Theoretically, the high PD-L1 expression observed in cells with *KRAS* mutation results from activation of ERK, which mediates upregulation of PD-L1 [95]; however, the efficacy of ICIs in treating NSCLC tumors harboring *EGFR* mutations is low, indicating that patients with *KRAS* but not *EGFR* mutations are likely to have a good response to ICIs. 

Mutations in genes that encode components of the mammalian switch/sucrose non-fermentable *(mSWI/SNF)* complex, which are occasionally identified in HCC, [91,96,97] appear to be associated with the efficacy of ICIs. Li et al. [98] reported that *ARID1A* mutation in gastrointestinal cancers is associated with TMB-H and high PD-L1 expression. The authors posited that *ARID1A* mutation may have utility as a biomarker for identifying patients with gastrointestinal cancer patients that are likely to respond to immunotherapy. A further study [67] based on WES data from 249 patients with solid cancers who received immunotherapy demonstrated that loss of *PBRM1* is correlated with response to anti-PD 1 or anti-PD-L1 therapy. However, in a large database [99] of 1544 patients with solid cancers (848 from Memorial Sloan Kettering Cancer Center and 676 from Dana-Farber Cancer Center) treated with immunotherapy, loss of function (LOF) in genes related to mSWI/SNF (*ARID1A*, *ARID1B*, *SMARCA4*, *SMARCB1*, *PBRM1*, and *ARID2*) were not correlated with ICI response. The conflicting results regarding the associations between immunotherapy response and mutations or LOF in genes related to the SWI/SNF complex appear to require further studies.

Ross et al. [100] conducted comprehensive genomic profiling in patients with NSCLC and reported that *MET* and *BRAF* alterations were associated with an increased duration of ICI treatment regardless of TMB value. Interestingly, MET overexpression is reportedly associated with poor prognosis in HCC [101]. 

### 2.3. Biomarkers Related to the Tumor Microenvironment 

#### Cytokines and Chemokines

Immune cells within the microenvironment can respond to signals received through their inherent receptors with their own protein-based language that will influence the cell itself and other cells throughout the organism. The language of cytokines is critical in this communication. Cytokines are small soluble factors with pleiotropic functions that are released by many cell types according to gene expression patterns. Types of human cytokines include interleukins, interferons, tumor necrosis factor (TNF), transforming growth factor-β (TGF-β), and other miscellaneous hematopoietins. [102] These cytokines have been evaluated as biomarkers of response to immunotherapy in various cancers. 

The binding between PD-1 and PD-L1 leads to immune system exhaustion and increased numbers of Tregs [103]. PD-L1 expression varies depending on local concentrations of pro-inflammatory cytokines [104]. IFN-γ is one of the most actively studied inflammatory cytokines. Increased IFN-γ levels are reportedly associated with a good response to immunotherapy [105,106,107]. In addition, TNF-α, [105,108,109] IL-6, [106,109,110,111] IL-8, [105,112,113,114,115] and TGF-β [18,116,117,118] are reported to have utility as predictive biomarkers for cancer immunotherapy. TGF-β is involved in immunosuppression within the TME and tumor immune evasion [119]. The inhibitory immunological function of Tregs is a major obstacle to eliciting an effective anti-tumor response, and Treg activation is modulated by the TGF-β pathway [120]. The TGF-β signaling pathway is activated at the transcriptional level in most HCCs [121,122], with a strong association demonstrated between the TGF-β signature and the exhausted immune signature in HCC [123]. In a Phase II study [116] of pembrolizumab for HCC by Lynn et al., high baseline plasma TGF-B levels (≥200 pg/mL) were significantly correlated with poor treatment outcomes after pembrolizumab treatment. In this study, there was no association between PD-L1 expression and pembrolizumab response. IL-6 is an inflammatory cytokine that has also been studied as a biomarker in HCC. Shakiba et al. [124] demonstrated significantly higher serum IL-6 levels in patients with HCC compared to healthy controls in a meta-analysis. Real-world data have revealed that high baseline IL-6 levels are associated with poor prognosis in patients with HCC treated with atezolizumab and bevacizumab [125]. Similarly, IL-27, an anti-inflammatory cytokine, has the potential to be a predictive biomarker. Active IL-27 receptor signaling was previously shown to reduce the expression of IL-6 and other inflammatory cytokines deemed pro-tumorigenic in HCC [126] and in another study, IL-27 was shown to have a similar function as IFN-γ and tends to be inhibited by IL-6 [127]. These results suggest that IL-27 might have a role as a predictive biomarker of immunotherapy for HCC. Turan Aghayev et al. also reported that elevated *IL27RA* mRNA expression correlated with poor survival since the treatment in Korean, Chinese, and TCGA HCC cohorts [128]. 

The chemokine, subdivided into four main classes depending on the location of the first two cysteine resides in their amino acid sequence: CC, CXC, C, and CX3C, is the largest subfamily of low molecular weight chemotactic cytokine [129]. Chemokines directly and indirectly affect tumor immunity, inflammatory response, proliferation, invasion, and metastasis via modulation of various signaling pathways including the JAK/STAT, PI3K/Akt, ERK1/2 MAPK, and Wnt/β-catenin pathways among others [130,131]. Chemokines related to HCC have been actively studied, and most chemokines have been confirmed to play important roles in tumor development and survival, including neovascularization, renewal, apoptosis, invasion, and metastasis in HCC [132,133,134,135,136,137,138,139,140,141,142,143,144,145,146,147,148,149,150,151,152,153]. Lin et al. posited that CXCLs are potential therapeutic targets for regulating anti-cancer immunity in HCC and may have utility as prognostic markers of response to immunotherapy [154]. Gu et al. also suggested that CCL14 may be a potential independent prognostic biomarker for HCC [141]. Previous studies evaluating chemokines as predictive biomarkers of the response to immunotherapy have been conducted using gene signatures comprising multiple chemokines (predominantly immune-related genes such as IFN-γ) rather than specific chemokines alone. The gene signatures with utility as predictive biomarkers are described in the following section. Studies on the effect of chemokines on tumor immunity in HCC and their potential as targets for immune modulation indicate that chemokines may have utility as predictive biomarkers of response to immunotherapy in HCC.

### 2.4. Tumor Infiltrating Lymphocyte (TIL)

TILs are defined as all lymphocytic substances within or around tumor cells and generally refer to CD4+ and CD8+ T cells. TILs are considered to be associated with a critical role in the anti-tumor immune response [155]. A significant association between high baseline TILs density and ICIs response has already been studied in RCC, CRC, NSCLC, and breast cancer [156,157,158,159,160,161]. In HCC, increased TIL was associated with a good response to ICI treatment. In the Checkmate 040 trial, increased CD3+/CD8+ tumor-infiltrating T cells of nivolumab-treated HCC patients showed a trend toward survival prolongation, but was not significant (*p* = 0.08) [162]. In another study of 32 HCC patients who underwent radiofrequency ablation or chemoablation with an injection of tremelimumab, the increase in CD8+ T cells identified by a 6-week tumor biopsy was associated with improved survival time [163]. Ng et al. also reported that high intra-tumoral CD38+ cells identified by immunochemistry were associated with a good response to ICI [164].

Clinical use of TILs as a biomarker to predict immunotherapy response in HCC seems challenging in the near future. Standardization and validation of test methods, test timing, and test interpretation should be needed. Nevertheless, since the role of TIL in the adaptive immune resistance mechanism is as essential as PD-L1, so it has high potential as a predictive biomarker of immunotherapy. 

## 3. Circulating Biomarkers

### 3.1. Circulating Tumor DNA and Circulating Tumor Cells

As molecular biology technology develops, interest in the use of liquid biopsies is also increasing due to a rising demand for non-invasive methods of obtaining genomic information from tumor cells. Accordingly, the evaluation of circulating tumor DNA (ctDNA) and circulating tumor cells (CTCs), also known as liquid biopsy, has been widely studied in recent years. ctDNAs are cell-free materials released by tumor cells into the bloodstream following tumor cell apoptosis or necrosis [165]. In HCC, ctDNA levels are correlated with tumor size, extrahepatic spread, and vascular invasion [166]. ctDNA levels are correlated with microvascular invasion and predict tumor recurrence of HCC. Franses et al. showed that the quality of genetic information in ctDNA is just as valuable as that in tissues. They performed ctDNA profiling using commercially available NGS assays in 136 patients with unresectable HCC from four cancer centers. In 28 patients, blood TMB (bTMB) levels were approximately three-fold higher than tissue TMB (tTMB) levels [167]. Qualitative analysis of somatic mutations in HCC-derived ctDNA has detected several oncogenes and tumor suppressor genes including *RAS*, *TERT*, *TP53*, *PTEN*, *ARID2,* and *CTNNB1* that are consistent with the results of tissue analyses in 63% of cases [168,169]. Fu et al. [170] investigated ctDNA in preoperative blood samples from 258 HCC patients who underwent curative liver resection. The number of gene alterations detected in ctDNA was associated with early tumor relapse. In this study, patients with *FAT1*, or *LRP1b* variants but without *TP53* variants had worse PFS following treatment with lenvatinib combined with ICIs. According to the NORTE STUDY group, baseline CXCL9 levels measured using a cytokine array of ctDNA were significantly lower in patients with early disease progression following treatment with atezolizumab and bevacizumab [171]. In a phase II study [172] of camrelizumab plus afatinib in HCC, ctDNA has utility in predicting pathologic response and relapse following treatment. A Japanese study investigated the potential role of cfDNA/ctDNA as biomarkers for predicting treatment response in patients with unresectable HCC who had been treated with atezolizumab and bevacizumab. High pre-treatment cfDNA levels were associated with a lower response rate and shorter PFS and OS. Further, the presence of a *TERT* mutation and a serum AFP levels ≥400 ng/mL were independent predictors of poor OS after treatment with atezolizumab combined with bevacizumab [173]. 

Circulating tumor cells are nucleated cells released into the bloodstream from tumor cells. The detection of CTCs remains challenging as CTCs are present in the blood at low concentrations and there is no standardization of testing methods [174]. Nevertheless, CTCs are considered attractive biomarkers as they have the characteristics of tumor cells. Chen et al. [175] reported that CTCs were detected in 95% of 195 patients with HCC, with a median number of 6 CTC in each 5 mL blood sample. The number of CTCs was reported to be correlated with disease stage (BCLC), metastasis, and serum AFP levels. The simple number of CTCs has been used to predict prognosis, including disease status or recurrence after surgery, in HCC [176,177,178]. Winograd et al. [179] confirmed that CTCs express immune checkpoints including PD-L1, PD-L2, and CTLA-4. PD-L1 expression in CTCs may have utility in predicting immunotherapy response in HCC [180]. However, the detection of CTCs remains challenging as CTCs are present at low concentrations in blood samples, and different methods may enrich different CTC populations, thereby affecting PD-L1 measurements. Similar to other novel biomarkers, there is an urgent need for the standardization of methods for quantifying CTCs. Prospective clinical trials of liquid biopsies for predicting the efficacy of ICIs are encouraged. 

### 3.2. Serum Alpha-Fetoprotein (AFP) and CRP

AFP is the most widely studied and used biomarker in HCC. In vitro, AFP has been proven to have an oncogenic effect by regulating TNF cytotoxicity [181], suppressing NK cell activity, [182] and promoting tumor growth by reducing levels of FAS-associated death domain protein (FADD) [183]. Recent phase 3 studies for ICIs have reported somewhat conflicting results about the predictive role of serum AFP levels. The IMbrave150 study reported [5] in a subset of patients with serum AFP levels <400 ng/mL, immunotherapy was associated with longer survival, while and the HIMALAYA [184] and CHECKMATE 459 study [6] reported in a subset of patients with serum AFP levels ≥400 ng/mL, immunotherapy was associated with longer survival compared to control arm group. However, post-treatment AFP values appear to be consistently associated with ICI response. The measurement of serum AFP levels at six weeks after initiating treatment is a potential surrogate biomarker of prognosis in patients with HCC receiving atezolizumab and bevacizumab. Zhu et al. investigated the relationship between changes in serum AFP levels and response to treatment in patients enrolled in the GO30140 and IMbrave150 studies. Based on a ≥75% decrease in serum AFP levels at six weeks after initial treatment, the sensitivity for discriminating between responders and non-responders was 0.59 with a specificity of 0.86 in the IMbrave150 study, while serum AFP levels using the same cut-off value had a sensitivity of 0.71 and a specificity of 0.91 in the GO30140 study. Lower serum AFP levels are reportedly associated with longer OS and PFS in patients with HCC, particularly in patients with HBV-related HCC [185]. Even in real-world data from patients with HCC treated with atezolizumab and bevacizumab, early changes (three weeks after treatment) in serum AFP levels were significantly associated with an objective response to treatment. An AFP ratio (AFP levels after treatment to baseline AFP levels) of 1.4 or higher three weeks after the initiation of treatment may be an early predictor of refractoriness to atezolizumab plus bevacizumab [186]. 

Serum CRP levels may also predict response to PD-1 inhibition. Zhang et al. reported baseline serum CRP and AFP levels may have the potential as predictors the efficacy of PD-1 inhibitors in HCC [187]. A Japanese study also demonstrated that serum AFP and CRP levels are associated with the efficacy of immunotherapy. The CRAFITY score, composed of serum AFP, and CRP levels, reportedly has utility in predicting the treatment outcomes and side effects of immunotherapy [188]. In a previous study conducted in Europe, the CRAFITY score had utility in predicting radiologic response and survival after immunotherapy [189]. 

### 3.3. Neutrophil-to-Lymphocyte Ratio (NLR) and Platelet-to-Lymphocyte Ratio (PLR) 

Circulating blood components such as platelets, granulocytes, and neutrophils are involved in tumor growth and metastasis and play a role as a pool of vascular endothelial growth factor (VEGF). In various diseases, the NLR and PLR are used as inflammatory markers and are actively studied in HCC. Elevated neutrophil and platelet count can result in elevated circulating VEGF levels and is associated with poor prognosis [190,191]. Huang et al. [192] investigated the prognostic value of blood biomarkers in 100 patients with HBV-induced HCC treated with PD-1 inhibitors. In this study, a high systemic immune inflammation index, high platelet-to-lymphocyte ratio (PLR), high neutrophil-to-lymphocyte ratio (NLR), and low lymphocyte-to-monocyte ratio were correlated with decreased OS and PFS. In a separate retrospective study [193] of 103 patients with HCC treated with nivolumab, post-treatment NLR and PLR were significantly lower in patients with PR or CR compared to patients with stable disease or PD. Post-treatment NLR and PLR were significantly associated with overall survival. NLR was also identified as a significant prognostic biomarker in three multicenter retrospective real-world studies of patients with HCC treated with atezolizumab/bevacizumab in East Asia. In a Korean study [194], the authors reported that a high baseline des-gamma-carboxy prothrombin level (≥186 mAU/mL), NLR ≥ 2.5, and a decrease in NLR ≥ 10% at first response may be useful prognostic predictors for OS and PFS. Similarly in a Japanese study, [195] high baseline NLR (>3) was significantly associated with poor survival in patients with HCC treated with atezolizumab/bevacizumab. In a Chinese study, patients with a baseline NLR ≥ 5 had significantly shorter OS and PFS compared to patients with an NLR < 5 [196].

## 4. Host-Related Biomarkers

### 4.1. Etiology

Worldwide, approximately 13% of cases of cancer are associated with infections. The four most important infectious agents associated with cancer are *Helicobacter pylori*, human papillomavirus (HPV), hepatitis B virus (HBV), and hepatitis C virus (HCV), which together account for more than 90% of infection-related cancers [197]. Viral infections are estimated to contribute to the development of 15–20% of human cancers. HPV-related head and neck cancers have a good treatment response and favorable prognosis. Viral-associated cancers have distinct biological and clinical features compared to other tumor types [198]. HCC is strongly linked to viral infection, with approximately 54% of cases of HCC attributed to HBV infection (which affects 400 million individuals globally), while 31% can be attributed to HCV infection (which affects 170 million individuals globally) [199]. Liver cirrhosis, which is associated with an increased risk of HCC, occurs under the influence of inflammatory cytokines [200,201]. According to a study by Beudeker et al., patients with HBV-related liver cirrhosis and HCC had the greatest upregulation of pro-inflammatory mediators compared to patients with cirrhosis due to HCV, alcohol-related liver disease, or non-alcoholic fatty liver disease (NAFLD) [202]. Non-alcoholic steatohepatitis (NASH) and NAFLD are representative causes of non-viral HCC. 

Pfister et al. confirmed the unfavorable effects of anti-PD-1 treatment on NASH in experimental mice models, providing evidence of the tissue-damaging role of CD8+PD-1+ T lymphocytes [203]. Several studies have reported inflammatory responses according to HCC etiology may represent a biomarker for predicting the response of immunotherapy; however, the results are somewhat controversial. The results of two recent meta-analyses revealed no significant difference in response to immunotherapy between patients with viral-associated HCC and non-viral HCC, with a similar response rate observed between patients with HBV and HCV infection [204,205]. Recent phase III studies [6,44,202,206,207] of ICI have demonstrated that immunotherapy tends to be more effective in cases of viral HCC. The ORR in this study of 27% with atezolizumab/bevacizumab, 12% with nivolumab, 19% with durvalumab, and 18% with durvalumab plus tremelimumab in patients with non-viral HCC compared with 32%, 19%, 14.3%, and 21.3%, respectively, for HBV-associated HCC and 30%, 17%, 22.4%, and 35.5%, respectively, for HCV-associated HCC suggest that immunotherapy has similar efficacy between non-viral HCC and viral HCC. The level of evidence for the lower efficacy of ICI treatment in non-viral HCC is very low as large-scale clinical trials have failed to produce concordant results. 

### 4.2. Performance Status and Liver Function 

As the number of elderly cancer patients increases along with the prolonged life expectancy, the number of patients with poor performance status also increases. Accordingly, interest in cancer management in these vulnerable groups has been also increasing. Performance status is one of the most reliable indicators for predicting cancer prognosis. A score greater than 2 on the ECOG performance status scale is generally accepted as a relative contraindication to systemic treatment including cytotoxic chemotherapy and treatment with immune checkpoint inhibitors. In most clinical trials, eligibility criteria require a performance status based on an ECOG score of at least 2. Even in patients with HCC, performance status plays an important role in determining treatment plans. The BCLC (Barcelona Clinic Liver Cancer) staging system, which provides a guide for first-line treatment of HCC, consists of disease extension, liver function, and performance status. According to the treatment recommendations in the BCLC staging system, BCLC-C, and BCLC-D are classified according to liver function and performance status. Systemic treatments such as atezolizumab/bevacizumab are recommended for patients with BCLC-C HCC, and the best supportive care is recommended for patients with BCLC-D HCC [208]. Scheiner et al. proposed a predictive scoring system using CRP and AFP for immunotherapy in HCC [189] and reported ECOG and Child–Pugh Class as independent prognostic factors related to OS in patients with HCC receiving immunotherapy after multivariate Cox regression analysis.

The ALBI (albumin-bilirubin) score was developed to evaluate liver function more objectively and simply by excluding subjective factors such as ascites and encephalopathy that are included in the Child–Pugh score. Accordingly, the ALBI score is also expected to predict the response to immunotherapy [209]. According to the Imbrave150 exploratory analysis [210], the response to atezolizumab/bevacizumab was better than the response to sorafenib in patients with ALBI grade 1 compared to patients with ALBI grade 2. A subgroup analysis of the HIMALAYA trial [7] demonstrated that durvalumab/tremelimumab was superior to sorafenib in patients with ALBI grades 1 and 2; however, this difference did not reach statistical significance. In real-world data [211], baseline ABLI is considered an independent predictive biomarker in patients with HCC treated with immunotherapy.

### 4.3. Disease Status and Tumor Burden

In clinical practice, HCC treatment is predominantly determined based on BCLC staging [208]. Systemic treatment including immunotherapy is recommended for BCLC-C. For patients with BCLC-B, an intermediate stage of disease, TACE is the preferred option and immunotherapy is considered an appropriate option for those with a larger tumor burden. Most recent phase III studies [5,7,212,213,214,215] on immunotherapy in HCC have focused on patients with BCLC-B and BCLC-C HCC. However, immunotherapy was reported to be effective in patients with BCLC-C HCC but not BCLC-B HCC compared to a control group (predominantly comprising patients treated with sorafenib), which could be due to underpowered analysis with a smaller sample size [5,7].

Total tumor burden is another factor associated with response to immunotherapy. Preclinical data demonstrate that PD-1 blockade is more effective in mice bearing smaller lung squamous cell tumors [216], and PD-L1 blockade had a greater effect in mice with ovarian tumors at an early stage [217]. This negative correlation between response to PD-L1 blockade and total tumor volume has been studied in human lung cancer and melanoma as well as in animal models [218,219,220,221,222]. Larger tumors tend to be more immunosuppressive at both the local and systemic levels compared to smaller tumors. Myeloid-derived suppressor cells, tumor-associated macrophages, and regulatory T cells (Tregs) have been shown to increasingly infiltrate the TME as tumors progress in preclinical studies [223,224]. In addition to the increase in immunosuppressive cellular components in TMEs, cytokine production is distorted to a more suppressive profile in large tumors compared to small tumors. Levels of TGF-β, which has anti-tumor effects in early-stage cancer but tumor–promoting effects in late-stage cancer, IL-10, and nitric oxide synthase 2 (NOS2) increase as tumors progress [225,226]. In a recently published Korean study reporting real-world data [227], the nivolumab response was significantly correlated with primary tumor size in 261 patients with advanced HCC. In addition, the authors reported decreasing responses to nivolumab in the order of intrahepatic tumors (ORR, 10.1%) followed by metastatic tumors in the lung (ORR, 24.2%) and LN (ORR, 37.1%). The above results are supported by a previous study demonstrating that sites of HCC metastasis have altered pathological features [228] and Tregs have distinctive functions through different mediators in other organs [229].

### 4.4. Gut Microbiome

The gut microbiome, which is critical for the development and regulation of innate and adaptive immunity, influences other organs including the brain, liver, and pancreas, and the development of various diseases including obesity [230], diabetes [231], and cancers [232]. In 2015, the relationship between the gut microbiome and the effect of immunotherapy was reported for the first time in preclinical mouse studies. Tumor growth, spontaneous anti-tumor immunity, and the efficacy of immune checkpoint inhibitors differ in mice depending on the composition of the gut microbiota [233,234]. Data from these preclinical studies indicate that inter-individual heterogeneity of immunotherapy efficacy may be partially caused by the gut microbiome, and studies on this effect are currently being conducted. An early mouse study [233] reported that the *Bacteroides* genus is associated with good anti-CTLA-4 response. However, the *Bacteroides* genus is reportedly associated with poor response to immunotherapy in humans [235]. *Bifidobacterium longum*, *Dorea formicigenerans, Collinsella aerofaciens, Alistipes putredinis*, and *Prevotella copri* are reportedly enriched in responders to anti-PD 1 treatment in melanoma [236] and non-small-cell lung cancer [237], with *Ruminococcus obeum*, and *Roseburia intestinalis* found to be more abundant in non-responders. Interestingly, the microbiome observed in responders was also associated with frequent immune-related colitis. 

The use of antibiotics can lead to changes in the composition and function of the gut microbiome which may reduce microbial diversity and adversely affect the immune response. Indeed, several studies [238,239,240,241] have reported that changes in the gut microbiome following the use of antibiotics can have negative impacts on responses to immunotherapy. The importance of gut microbial diversity has also been studied in HCC. Zheng et al. [239] reported that fecal samples from patients responding to immunotherapy had higher taxa richness and more gene counts than those of non-responders. In this study, both responders, and non-responders had similar microbial composition to healthy people before treatment. However, responders still had a stable microbiome while non-responders had increased proteobacteria after treatment which became dominant over time. The results of this study indicate that gut microbial diversity and stability are associated with the response to immunotherapy. 

Microbial signatures can be used to stratify patients according to the likelihood of treatment response or toxicity. Modulating the gut microbiota may represent a potential treatment strategy for cancer; however, these approaches are likely to require adaption depending on the cancer type and therapeutic drug type. Large-scale studies monitoring sequential changes in the gut microbiome following the administration of immunotherapies are required to determine the utility of microbial signatures in predicting responses to immunotherapy.

## 5. Conclusions

The advent of immunotherapy with immune checkpoint inhibitors has shed new light on HCC treatments. However, immunotherapy has only recently been approved as a standard frontline therapy. Accordingly, there is a lack of studies on biomarkers that are able to predict the efficacy of immunotherapy in HCC. No reliable biomarkers with utility in predicting responses to immunotherapy have been identified to date as only studies on tissue PD-L1 expression and TMB has been published in addition to the results of exploratory analyses in phase III studies (Table 1.) However, increased real-world data from patients treated with ICIs are expected which may facilitate the development of more precise and accurate predictive biomarkers that improve personalized cancer therapy. Liquid biopsy and microbiome might have a role in understanding TME and inflammation, which has a strong link with the immunotherapy response of HCC.

In addition, although not mentioned in the text there are more potential candidates as a predictive biomarker for immunotherapy in HCC. Sex and age are the most basic information that shows distinguishing features immunologically. On average, women have stronger innate and adaptive immune responses than men [244], Therefore, the benefit from immunotherapy is also expected to be small. In meta-analyses of solid cancers, survival time after immunotherapy was revealed to be longer in male patients than in female patients [245,246]. As age increases, there is a tendency to experience various side effects and more severe toxicity after immunotherapy. Indeed, in the recent phase III trials of ICI for HCC [247], an increasing population over 65 was associated with lower ORR and reduced survival. Smoking also causes chronic inflammation, which can contribute to alterations in immune response. A strong association between smoking and TMB-H has already been demonstrated in NSCLC [248,249], and Wang et al. [250] found that smoking in HBV-related HCC affects the immune response through viral activation.

In conclusion, a combinatory approach that considers the intrinsic feature of the tumor, the peritumoral microenvironment, the immune system, host factors, and their clinical and molecular analyzes is likely required for the prediction of immunotherapy response in HCC. Moreover, considering the constantly changing TME and diverse tumor biology of HCC, further research on personalized biomarkers that enable continuous monitoring in a non-invasive, and cost-effective way is needed. 

## Figures and Tables

**Figure 1 ijms-24-07640-f001:**
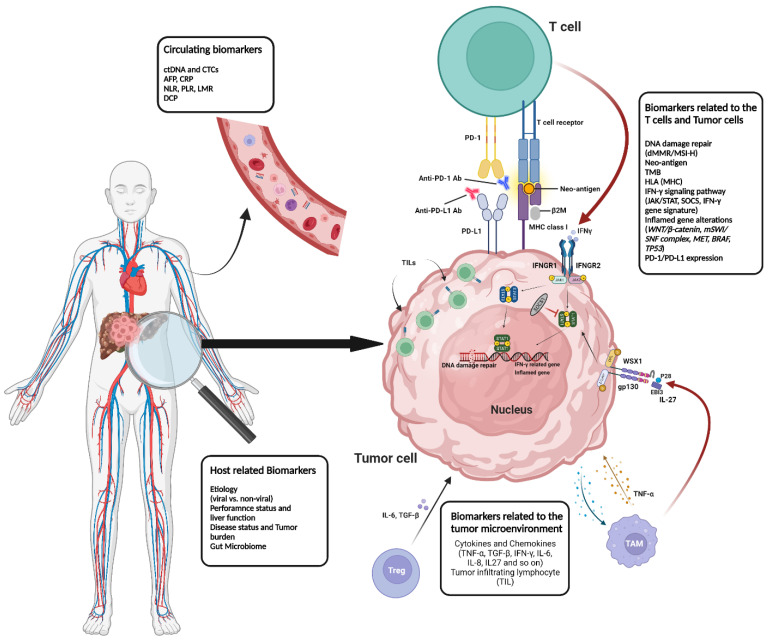
Schematic figure representing predictive biomarkers of immunotherapy in HCC.

**Table 1 ijms-24-07640-t001:** HR of subgroups related to predict OS in recent phase 3 studies for immunotherapy in HCC (Immunotherapy versus sorafenib).

	IMBRAVE150 [44]	CHECKMATE 459 [6,242]	HIMALAYA [45,207,243] (STRIDE/Durvalumab)
Age ≥ 65	0.69 (0.49–0.98)	0.88 (0.68–1.12)	0.73 (0.58–0.93)/0.83 (0.66–1.06)
Male	0.64 (0.49–0.83)		0.73 (0.61–0.88)/0.86 (0.72–1.03)
Asian	0.62 (0.42–0.93)	0.74 (0.56–0.98)	0.68 (0.52–0.89)/0.83 (0.64–1.06)
ECOG 1	0.56 (0.39–0.80)		0.74 (0.57–0.95)/0.85 (0.66–1.10)
BCLC C	0.63 (0.48–0.82)	0.78 (0.65–0.95)	0.76 (0.63–0.91)/0.86 (0.72–1.03)
AFP			
< 400 ng/mL	0.58 (0.42–0.81)	0.98 (0.78–1.24)	0.82 (0.63–1.05)/0.78 (0.61–1.01)
≥ 400 ng/mL	0.77 (0.53–1.12)	0.67 (0.51–0.88)	0.64 (0.45–0.91)/0.73 (0.53–1.03)
MVI: No	0.66 (0.47–0.92)		0.77 (0.63–0.93)/0.87 (0.72–1.05)
EH spread: Yes	0.60 (0.44–0.81)		0.67 (0.53–0.84)/0.79 (0.64–0.98)
MVI and/orEH spread: YES	0.64 (0.49–0.85)	0.74 (0.61–0.90)	0.73 (0.59–0.89)/0.82 (0.67–1.00)
HBV	0.58 (0.40–0.83)	0.79 (0.59–1.07)	0.64 (0.48–0.86)/0.78 (0.58–1.04)
HCV	0.43 (0.25–0.73)	0.72 (0.51–1.02)	1.06 (0.76–1.49)/1.05 (0.75–1.48)
PD-L1 expression			
TC or IC ≥ 1% (SP263)	0.52 (0.32–0.87)		
TC ≥ 1% (28-8, Dako)		0.80 (0.54–1.19)	
TC ≥ 1% (SP263)			0.85 (0.65–1.11)/0.87 (0.66–1.13)

## Data Availability

Not applicable.

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
