# Peer review of "Predictive Biomarkers for Immune-Checkpoint Inhibitor Treatment Response in Patients with Hepatocellular Carcinoma"

_ijms, 2023, doi:10.3390/ijms24087640_

Round 1
Reviewer 1 Report
In this review article, the authors summarized potential biomarkers for prediction of treatment response to immune-checkpoint inhibitor (ICI)-based immunotherapy in HCC.
There are four categories including 1. Biomarkers related to the T cells and tumor cells, 2. Biomarkers related to the tumor microenvironment, 3. Circulating biomarkers and 4. Host related biomarkers.
In my opinion, the authors need to explain the mode of action of ICI in HCC at the beginning of introduction and then discuss potential biomarkers.
Specific comments:
1. In the first part, the authors discuss the impact of DNA damage repair (DDR) pathway in HCC. This part is little confusing and needs to revise. The authors need to discuss the impact of dMMR/MSI-H (this word should be explained) on HCC development and the treatment response to ICI. Notably, how is the impact of dMMR/MSI-H on PD-L1 expression?
2. In the first part, the authors mentioned the tumor mutational burden (TMB), which is also discussed later in the independent part. The authors need to consider the rearrangement of each part.
3. Biomarkers related to the tumor microenvironment may be combined with circulating biomarkers.
Author Response
In this review article, the authors summarized potential biomarkers for prediction of treatment response to immune-checkpoint inhibitor (ICI)-based immunotherapy in HCC.
There are four categories including 1. Biomarkers related to the T cells and tumor cells, 2. Biomarkers related to the tumor microenvironment, 3. Circulating biomarkers and 4. Host related biomarkers.
- We appreciate the time and effort you have dedicated to providing insightful feedback on ways to strengthen our review. To facilitate your review of our revisions, the following is a point-by point response to the comments.
In my opinion, the authors need to explain the mode of action of ICI in HCC at the beginning of introduction and then discuss potential biomarkers.
- Thank you and we agree with your opinion. We added phrases for action mechanism of ICIs in HCC In the beginning of introduction with references.
You can see at Lines 10-23, Page 4, after revision
The immune system can recognize foreign cells based on the proteins present on the cell surface and has ability to eliminate different from our own body such as viruses, bacteria and malignancies. In this process, checkpoint proteins are to limit autoimmune damage to normal tissue by preventing T cell activation. HCC and other tumors use this mechanism to evade immune responses by expressing ligands in tumor cell surface. In addition, tumor cells promote immune evasion through interfering with the recognition of tumor antigen presentation or generating of immunosuppressive tumor microenvironment. immune checkpoint inhibitors block the interaction between checkpoint proteins and their ligands, thereby preventing the inactivation of T cell function. Chronic viral infections such as hepatitis B and hepatitis C, which are the main causes of HCC, promote chronic inflammation of the liver, and in patients with chronic inflammatory liver disease, PD-1 overexpression in lymphocyte and PD-L1/PD-L2 overexpression in stromal cells (Kupffer cell, liver sinusoidal endothelial cells) are observed. These upregulation of checkpoint proteins suggests that ICIs may be effective in HCC. Actually, the use of immunotherapy alone or in combination with targeted agents results in improved survival with a durable response and has become the new standard of care with several successful phase 3 studies published since 2020.
Specific comments:
In the first part, the authors discuss the impact of DNA damage repair (DDR) pathway in HCC. This part is little confusing and needs to revise. The authors need to discuss the impact of dMMR/MSI-H (this word should be explained) on HCC development and the treatment response to ICI. Notably, how is the impact of dMMR/MSI-H on PD-L1 expression?
- Thank you for your suggestion. As the reviewer commented, we agreed that the first part, the DNA damage repair pathway section, lacked discussion and mention about dMMR/MSI-H. In fact, unlike other tumors, HCC has a very low frequency of patients with PD-L1 overexpression or dMMR/MSI-H, so there are insufficient studies that can be used as a basis for predicting the response of ICIs. After discussions by the authors, the contents about dMMR/MSI-H were drastically reduced in the drafting process. In the revised manuscript, we added the explanation and significance of dMMR/MSI-H in HCC.
You can see at Lines 18-22, Page 6, after revision
The dramatic response of ICI shown by patients in the dMMR/MSI-H group in other tumors led to expectations that those of dMMR/MSI-H in HCC would play the similar role, but contrary to expectation, dMMR/MSI-H in HCC is limited in its use as a predictive biomarker for immunotherapy due to their low prevalence of less of than 3%.
In the first part, the authors mentioned the tumor mutational burden (TMB), which is also discussed later in the independent part. The authors need to consider the rearrangement of each part.
- Thank you for the comment. We agree that TMB was frequently mentioned in our review. TMB can be seen 60 times in our manuscript. The first was in the “abstract”, the next 6 times in the “DNA damage repair pathway (dMMR/MSI-H)”, 38 times in the TMB part, 4 times in the “alterations in genes related to immune function” part, for explanation of blood TMB, it was mentioned 4 times in “circulating biomarker”, 3 times in “Sex and age”, additional 3 times in “smoking” part, and the last 1 in “Conclusion”. The authors think that TMB is an important predictive biomarker for immunotherapy in HCC. TMB was frequently mentioned to explain how other biomarkers relate to TMB-H.
Biomarkers related to the tumor microenvironment may be combined with circulating biomarkers.
- We fully agreed with your comment. Chemokine and cytokine are biomarkers related to the TME, but they are also representative soluble proteomic biomarkers, so they are applicable to circulating biomarkers. However, since the authors believe that the TME is very important in immune checkpoint inhibition, instead of combining it with the “circulating biomarkers” part suggested by reviewer, a new section of Tumor infiltrating lymphocyte (TILS) was added to the “Biomarkers related to the tumor microenvironment” part.
You can see at Lines 6-20, Page 15, after revision and revised figure 1.
Tumor Infiltrating Lymphocyte (TIL)
TILs are defined as all lymphocytic substances within or around tumor cells and generally referring to CD4+ and CD8+ T cells. TILs are considered to be associated with a critical role in the anti-tumor immune response. A significant association between high baseline TILs density and ICIs response has already been studied in RCC, CRC, NSCLC, and breast cancer. In HCC, increased TIL was associated with a good response to ICI treatment. In the Checkmate 040 trial, increased CD3+/CD8+ Tumor infiltrating T cells of nivolumab treated HCC patients showed a trend toward survival prolongation, but was not significant. (p=0.08) In another study of 32 HCC patients who underwent radiofrequency ablation or chemoablation with injection of tremelimumab, the increase in CD8+ T cells identified by 6-week tumor biopsy was associated with improved survival time. Ng et al. also reported that high intra-tumoral CD38+ cells identified by immunochemistry were associated with a good response to ICI.
Clinical use of TILs as a biomarker to predict immunotherapy response in HCC seems challenging in the near future. Standardization and validation of test methods, test timing, and test interpretation should be needed. Nevertheless, since the role of TIL in the adaptive immune resistance mechanism is as essential as PD-L1, so it has high potential as a predictive biomarker of immunotherapy. [The blockade of immune checkpoints in cancer immunotherapy
Reviewer 2 Report
Immune checkpoint inhibitor (ICI, including anti-programmed cell death [PD]-1/programmed death-ligand1 [PD-L1] antibodies and anti-CTLA-4 antibody)-based immunotherapy has presented a new milestone in the treatment of hepatocellular carcinoma (HCC). Based on the therapeutic effects of ICI alone, investigators have developed combined ICI treatments. Therefore, the development of biomarkers to predict the toxicity and treatment response in HCC patients receiving ICI is in urgent need. In this manuscript, the authors reviewed and discussed the current state of immunotherapy for HCC, the potentially predictive biomarker (including biomarkers related to the interaction between T cells and tumor cells, circulating biomarkers, and host-related biomarkers), and the future direction. The authors suggested that a combinatory approach that considers the intrinsic feature of tumor, the peri-tumoral microenvironment, the immune system, host factors, and their clinical and molecular analyzes is likely required for predicting immunotherapy response in HCC.
This is a comprehensive review on the potentially predictive biomarkers for ICI treatment response in patients with hepatocellular carcinoma. The manuscript is well prepared. The originality of this article is high. This review may provide useful information for the clinicians to treat the HCC patients with immune checkpoint inhibitors.
Author Response
Immune checkpoint inhibitor (ICI, including anti-programmed cell death [PD]-1/programmed death-ligand1 [PD-L1] antibodies and anti-CTLA-4 antibody)-based immunotherapy has presented a new milestone in the treatment of hepatocellular carcinoma (HCC). Based on the therapeutic effects of ICI alone, investigators have developed combined ICI treatments. Therefore, the development of biomarkers to predict the toxicity and treatment response in HCC patients receiving ICI is in urgent need. In this manuscript, the authors reviewed and discussed the current state of immunotherapy for HCC, the potentially predictive biomarker (including biomarkers related to the interaction between T cells and tumor cells, circulating biomarkers, and host-related biomarkers), and the future direction. The authors suggested that a combinatory approach that considers the intrinsic feature of tumor, the peri-tumoral microenvironment, the immune system, host factors, and their clinical and molecular analyzes is likely required for predicting immunotherapy response in HCC.
This is a comprehensive review on the potentially predictive biomarkers for ICI treatment response in patients with hepatocellular carcinoma. The manuscript is well prepared. The originality of this article is high. This review may provide useful information for the clinicians to treat the HCC patients with immune checkpoint inhibitors.
- The authors were deeply grateful for the reviewer`s generous and thoughtful comment.

Reviewer 3 Report
Thank you for the opportunity to review the article. The authors are commended for undertaking the challenge of bringing together the limited data and conflicting reports on prognostic and response biomarkers in HCC treated with ICI. As such, a review article on this topic is challenging.
My main suggesting would be that the likely significant influence of underlying cirrhosis etiology is buried deep within the article. I would recommend leading the body of the article with this overview, acknowledging that much of the available biomarker data is in patients with HBV-HCC. This will likely further complicate strategies to develop prognostic and response biomarkers, as many of these approaches may have etiological specificity.
The host-related biomarkers section breaks the flow of the review, specifically with regard to the biomarker-based focus. Specifically, the sections on sex / age, smoking, and early-response sections lack the depth of the other sections and could be excluded or collected and briefly summarized in a concluding section.
Author Response
Thank you for the opportunity to review the article. The authors are commended for undertaking the challenge of bringing together the limited data and conflicting reports on prognostic and response biomarkers in HCC treated with ICI. As such, a review article on this topic is challenging.
- We appreciate the time and effort you have dedicated to providing insightful feedback on ways to strengthen our paper. To facilitate your review of our revisions, the following is a point-by point response to the questions and comments.
My main suggesting would be that the likely significant influence of underlying cirrhosis etiology is buried deep within the article. I would recommend leading the body of the article with this overview, acknowledging that much of the available biomarker data is in patients with HBV-HCC. This will likely further complicate strategies to develop prognostic and response biomarkers, as many of these approaches may have etiological specificity.
- Thanks for your suggestion. As reviewer commented, we agreed that most studies on available biomarkers in HCC have been conducted for HBV-HCC. Recent phase 3 studies also included 60-70% of HCC patients with viral hepatitis (IMbrave 150: 48% of HBV and 22% of HCV, HIMALAYA: 31% of HBV and 27% of HCV). We’ll add statements about the underlying liver cirrhosis, the etiology of HCC, to the introduction.
You can see this phrase at Lines 3-6, Page 4, after revision
Repeated necrosis and regeneration of hepatocyte caused by chronic inflammation and damage gradually progresses to liver fibrosis and liver cirrhosis, eventually leading to HCC. Unfortunately, most patients with liver cirrhosis are asymptomatic, and so HCC is often diagnosed at advanced stages.
The host-related biomarkers section breaks the flow of the review, specifically with regard to the biomarker-based focus. Specifically, the sections on sex / age, smoking, and early-response sections lack the depth of the other sections and could be excluded or collected and briefly summarized in a concluding section.
- The authors agreed with the reviewer’s opinion that “host related biomarkers” hinder the flow of the review due to weak evidence in HCC. We’ll delete or reduce the content about “Sex and age”, “Smoking”, and “Early tumor response”. However, we think “etiology”, “Gut microbiome”, “Performance status” and “Liver function”, “Disease status”, and “Tumor burden” are accordance with potential predictive biomarker that should be considered at least once before immunotherapy, so the “Host related biomarkers” chapter was reduced but left. Actually, in the IMbrave 150 study and the HIMALAYA study, etiology, disease status, liver function, tumor burden and ECOG (performance status) were commonly used as stratification factors for balanced randomization.
You can go through the modified phrases and figure at Lines 11-21, Page 24, after revision
In addition, although not mentioned in the text there are more potential candidates as predictive biomarker for immunotherapy in HCC. Sex and age are the most basic information that show distinguishing features immunologically. On average, women have stronger innate and adaptive immune response than men, Therefore, the benefit from immunotherapy is also expected to be small. In meta-analyses of solid cancers, survival time after immunotherapy was revealed to be longer in male patients than in female patients. As age increases, there is a tendency to experience various side effects and more severe toxicity after immunotherapy. Indeed, in the recent phase III trials of ICI for HCC, an increasing population over 65 was associated with lower ORR and reduced survival. Smoking also causes chronic inflammation, which can contribute to alterations of immune response. A strong association between smoking and TMB-H has already been demonstrated in NSCLC, and Wang et al. found that smoking in HBV-related HCC affects the immune response through viral activation.
